# Enamel Evaluation after Debonding of Fixed Retention and Polishing Treatment with Three Different Methods

**DOI:** 10.3390/ma16062403

**Published:** 2023-03-17

**Authors:** Angelica Iglesias, Teresa Flores, Javier Moyano, Montserrat Artés, Nuria Botella, Javier Gil, Andreu Puigdollers

**Affiliations:** 1Department of Orthodontics, Universitat Internacional de Catalunya, 08195 Barcelona, Spain; 2Statistic Advisors Service, Universitat Internacional de Catalunya, 08195 Barcelona, Spain; 3Bioengineering Institute of Technology, Universitat Internacional de Catalunya, 08195 Barcelona, Spain

**Keywords:** dental debonding, retention, dental polishing, orthodontics

## Abstract

Lack of standardization of the retention phase has led to many studies of stability of movements and characteristic of retainers, disregarding the enamel repercussions of fixed retention on this phase. This study aimed to analyze different methods of enamel polishing after detachment of orthodontic retainers. Forty-five healthy premolars were divided into three groups according to the polishing bur after debonding, and four specimens without intervention were used as control. A 0.038 × 0.015 inches gold chain was bonded between the premolars and then removed. The adhesive remnant was removed with three types of burs according to the study groups (Group 1: white stone at high speed; Group 2: high-speed handpiece with a 30-blade tungsten carbide bur; Group 3: low-speed handpiece and a 30-blade tungsten bur). After debonding and polishing, all samples were analyzed with a confocal microscopy on surface roughness parameters: Sa: Arithmetic mean of the height of the surface. Rq: Square mean of the height of the surface and Sz: Maximum surface height. Mechanical tests were carried out to determine the bonding stress of the retention adhered to the teeth using an electromechanical testing machine. The adhesion stress was 8.23 MPa (±0.87). The quality of the refinement of the enamel after debonding is essential in order to preserve its integrity. The use of the 30-blade tungsten carbide bur provides a smooth enamel surface after polishing.

## 1. Introduction

The retention phase is the extended and long-lasting part of the orthodontic treatment, and it is a crucial part of orthodontic treatment. Its importance keeps increasing since patients look for a long-lasting “perfect” result, mainly for aesthetic reasons, even though some degree of relapse is always expected. For this reason, long-life retentions are more commonly advised every day by clinicians [1].

Many studies have analyzed the retention phase in terms of: stability, retention material, adhesion, clinicians and patients’ preference, and hygiene. Researchers have described the best characteristics for adhesion systems in fixed retainers, considering mechanical properties of the composite and wear resistance [2,3], while other authors focused on long-term stability of the tooth position [4] and periodontal health in comparison with control groups or removable retentions [5,6], but there is a lack of literature focused on the study of the consequences of retention in the enamel.

Usually, lingual retainers’ detachment is accidental and may be caused by an excessive force, adhesive material wear or retainer rupture. Any kind of rupture or need to repair the retention could cause alterations in the enamel due to the rupture in the adhesive interphase or the removal processes of remaining adhesive or retainer materials [7,8,9,10]. Selection of burs and rotary instruments will affect the ability to remove the remnant materials while minimizing the damage to the enamel structure.

Thanks to advanced microscopy technology and mineral property analysis techniques, the knowledge of enamel composition and its properties before and after adhesive treatments has been widely studied. The vast majority of studies are based on the vestibular surface because of the superior amount of vestibular appliances treatments (vestibular brackets and atachments for aligner therapy) in comparison with lingual appliances, but also due to an esthetical concern. However, on the lingual surface, more aggressive bonding techniques are often used given that this surface does not have an aesthetical importance and is rarely compared with the buccal side. An in vitro study using a Scanning Electron Microscope (SEM) found important differences between the two enamel surfaces. The lingual surface appears to be smoother, with smaller micropores. These interesting data are rarely discussed when adhesion protocols for retainers or lingual brackets are presented, and hence in the effect on the enamel after debonding [11].

The variability of the retention protocols and lack of studies on re-bonding effects and techniques for lingual retention results in no consensus with regard to the ideal adhesive removal method [12]. The various techniques include: hand instruments (pliers), rotatory instruments (high and low speed), sandblasting, ultrasound, and several bur and disc materials including: tungsten carbide burs, diamond burs, composite burs, rubber burs, and soft-lex discs [13,14,15].

The aim of this study is to analyze different methods of adhesive remnant removal techniques, after detachment of lingual retainers, in order to quantify the debonding stress of the retention and observe the repercussions of this removal techniques on the enamel, especially considering that long-life retention may very likely require one or several re-bonding procedures throughout life [16]. The null hypothesis is that there are no significant differences in enamel roughness among the different polishing methods.

## 2. Materials and Methods

### 2.1. Sample Size

A pilot study was conducted for an unpublished Master’s thesis that tested the methodology. Afterwards, sample size calculation was set to achieve a statistical power of 80% with a 5% significance level: taking a 2.5 standard deviation into account, resulting in 15 premolars per study group.

### 2.2. Sample Preparation

Forty-nine healthy premolars, extracted for orthodontical reasons in adolescents, were stored in 0.9% physiological saline and stored at 37 °C until sample preparation and testing. All samples were cleaned with gauze, removing all organic residues, and 4 samples were set aside to serve as control for untreated enamel. For the rest of the samples, bonding surface was abraded with a cup and fluoride-free prophylaxis paste (Zircate Phrofhy Paste, Dentsply Sirona, York, PA, USA) applied with a handpiece at a low speed for 10 s, posteriorly washed with water and dried with oil-free compressed air. Subsequently, the 45 samples were divided into 3 experimental groups of 15 specimens each, according to the polishing bur after debonding. All samples were mounted in a handmade silicon base (Hydrorise Model, Zhermack, Marl, Germany) in groups of three, in a position of alignment of the buccal surfaces.

### 2.3. Retention Bonding

A 0.038 × 0.015 inches gold chain (Reliance Ortho Prod. Inc., Itasca, IL, USA) was bonded between the three premolars of each block. The bonding process consisted of the application of acid etching with ortho-phosphoric acid 37% for 20 s and posterior rinse and dry with water and oil free compressed air. After acid etching preparation, the Transbond XT Adhesive Primer (3M Unitek, Monrovia, CA, USA) was rubbed in a thin layer for three seconds, followed by 20 s of light cured polymerization with a LED lamp (Bluephase, Ivoclar Vivadent AG, Schaan, Liechtenstein). The gold chain was placed with a cotton tweezer and Tansbond XT resin was applied over the surface of both the chain and tooth with posterior light cured polymerization for 20 s. The composition of fixed retention are showed in Table 1.

### 2.4. Debonding and Polishing Procedure

After all retentions were bonded, the debonding procedure was performed initially applying a lever pressure on the retainer, with a tweezer by the same operator. After all golden chains were detached, the removal of the remaining adhesive was performed by the same experienced operator, using a standarized technique based on the number of passes the bur made on the enamel surface for each tooth until total elimination of the composite remnants, corroborated by a visual criteria, ressembeling clinical conditions. Afterwards, three types of burs according to the three study groups were used to polish the enamel surface; the drills were positioned parallel to the long axis of the tooth, making lateral movements in the mesiodistal direction of the crown.

Three polishing systems were studied and presented in Table 2. Group 1 samples were polished with a high-speed handpiece with a white stone bur with constant water irrigation to remove all composite remnants and stored in 0.9% physiological saline at 37 °C.

For Group 2, the removal of the remnant adhesive was also performed by a high-speed handpiece with a 30-blade tungsten carbide bur with constant water irrigation and stored as Group 1.

Finally, Group 3 samples were polished with a low-speed hand piece with a 30-blade tungsten carbide bur, with constant water irrigation and posterior storage in saline water like samples in Groups 1 and 2.

### 2.5. Confocal Microscopy

After debonding and polishing processes, all samples were analyzed with a confocal microscopy to measure several rugosity parameters. A 5 square mm was outlined (with a bur) on the enamel surface to delimit the area of study, and the MCF Leica DCM3D (Leica microsystems, Wetzlar, Germany) was used to observe the surface roughness. Thirty-three planes of each surface at 12 µm between each plane were made to construct the 3D images and obatin the following parameters: Sa: Arithmetic mean of the height of the surface. Sq: Square mean of the height of the surface. Sz: Maximum surface height (Figure 1).

Image acquisition software included the LeicaSCAN DCM3D 3.2.3 and the LeicaMap 6.2.6561 image treatment software. The images were captured with a Leica HCX PL Fluotar objective at 10× magnification, with 0.30 numerical aperture, an X/Y optical resolution of 470 nm, a vertical resolution of <30 nm, and a confocal full resolution of 12.5 fps frequency.

### 2.6. SEM Evaluation

A qualitative study of the surface was performed on the samples by means of a scanning electron microscope. Two specimens of each group were stored in a vacuum desiccator for several days at room temperature. Before the observations, the samples were coated with a gold layer. Then, the specimens were analyzed with a scanning electron microscopy (SEM) (JEOL JSM 5410, Tokyo, Japan) operated at 10 kV, and 10 images per sample were obtained for analysis.

### 2.7. Mechanical Tests

The premolars were glued with the retention wire and, with the different treatments studied, were placed in a stainless steel mold using a high-strength resin. The specimens were placed with a high clamping into the jaws of the electromechanical mechanical testing machine. We ensured the immobility of the specimens for the mechanical tensile tests. Through the bridge of wire among the pads bonded to each tooth, the specimen was attached to the clamp with a high stiffness martensitic stainless steel wire (piano wire) AISI 314 of 0.5 mm diameter. The high stiffness of the wire eliminates the plastic deformation of the wire and therefore the results have a higher accuracy. From these mechanical tests, the adhesion strength can be determined as the force per unit area of the pad of composite resin with the retention wire to separate it from the tooth.

Tensile tests were performed on 10 samples of each of the treatments with an Adamel Lhomargy^®^ electromechanical machine model DY34 (Adamel Lhomargy SARL, Roissy en Brie, France) equipped with a 10 kN load cell. The tensile test was controlled with the Autotrac^®^ software (version: 3.0, Adamel Lhomargy SARL), and the test speed was 1 mm/min. The proper orientation of the specimen with the tensile axis is very important, and spherical plain bearings were used to allow perfect alignment of the specimen [2]. The correct orientation was ensured by means of a long-range microscope. The scheme of the mechanical tests can be seen in Figure 2.

### 2.8. Statistical Analysis

Numerical variables were described with mean and standard deviation, while the median was described within quartiles. Variables of Sq, Sa, and Sz were tested to probe normality by a Shapiro test, in which only the variable Sa followed the distribution.

These data called for the performance of a *t*-test for the Sa variable to obtain the difference with a trust interval of 95%. Meanwhile, for the other two variables, a Mann–Whitney test yielded the *p*-value data with a 95% trust interval.

Results were analyzed with a statistical software program (Statgraphics, Centurion XVIII, Warrenton, VA, USA). The level of statistical significance was set at *p* < 0.05.

## 3. Results

Results showed statistical significance in several parameters between the three groups when compared with the control. The 3D medium roughness (Sa) showed significant increased values for all the groups when compared with the control, with Group 1 representing the greater difference and the samples in Group 3 being the ones closer to the group control media (Table 3 and Figure 3).

For the Sq values, which are more sensitive to variations between peaks and valleys, the statistics showed significant differences with the control group only in the high-speed handpiece with a white stone group. Finally, in the Sz analysis of maximum surface height, no statistically significant differences were found in any of the groups; however, it followed the same pattern as the other two variables with Group 1 presenting the most augmented values and Group 3 with the values closest to the control group of untreated enamel (Table 4).

In the images obained by confocal microscopy (Figure 4, Figure 5 and Figure 6) and the SEM evaluation (Figure 7, Figure 8 and Figure 9), we can appreciate greater agression of the enamel surface on samples polished with a high-speed handpiece (Groups 1 and 2) when compared with the low-speed handpiece polishing group (Group 3).

The bond strength between the tooth and the composite pad and retention wire with the electromechanical testing machine resulted in being 8.23 MPa (±0.87).

## 4. Discussion

Enamel composition and its poor ability to restore itself once its structure has been damaged makes it vital to create protocols that produce the least possible iatrogenesis. The use of less aggressive burs for the removal of residual cement and the constant effort to minimize the great structural damage must be a critical concern to clinicians. According to Cardoso et al. [17], an optimal removal method must be able to remove the adhesive without compromising the enamel. The preservation of the enamel is of great importance in all temporarily bonding procedures and also in fixed retentions.

Many studies have analyzed the enamel characteristics after bracket debonding concerning enamel loss, roughness, shear bond strength, and adhesive remnant index [18,19]. However, no other study was found by the investigators with regard to these same parameters in lingual retentions. The differences between the two may rely on the differences between buccal and lingual enamel.

According to Brosh [20], the surface of the lingual enamel appears to be smoother in samples when compared with the buccal side; this may be a result of the constant contact with the tongue and closeness to the salivary glands’ foramens. This causes a macro-smoother pattern, smaller micro-pores, and a less pronounced wave-like appearance after conditioning that results in less mechanical interlocking, lower debonding strength, and higher tooth damage compared with the buccal side.

However, in our research, we acknowledged our limited ability to make comparisons with other studies given that the vast majority of them are evaluating data from the buccal surface of teeth; in addition, there is a greater limitation in obtaining samples from lower incisors, which are the most commonly used area for lingual fixed retention. Therefore, we decided to perform our analysis on the buccal surface of healthy extracted premolars, focusing our results on the effects of the polishing on enamel rather than the area of treatment.

The Atomic Force Microscopy (AFM) is a nanometric technique that allows us to evaluate the enamel without the need for a previous sample preparation that alters its structure. However, in the literature, we often find a greater amount of methodology that results in qualitative findings instead of quantitative data.

The statistically significant differences for the Sa in the three groups, compared with the control, indicate a greater rugosity in all samples after adhesive removal. This is in agreement with other authors [21,22]. This enamel aggression affects the esthetic properties, bacterial adhesion, and plaque formation of enamel by altering the pathogenic environment [23,24].

On the other hand, Shah et al. [14] found no statistically significant difference between groups after polishing, which may imply that a polishing protocol after debonding would be advisable to improve the enamel surface. However, in our study, the mean roughness of the enamel after polishing with tungsten carbide burs (both high- or low-speed) showed less roughness than with a white stone bur, while the maximun surface height for the three groups showed no statistically significant differences with the control group.

As we have yet to find a non-harmful method, researchers have examined the potential use of toothpastes containing surface (S)-pretreated glass-ionomer (PRG) filler [22,25], based on calcium phosphate [26], or a novel fluoride-containing bioactive glass [27] to develop remineralization protocols to recover enamel loss. New studies [26,27,28] have also inquired about the citotoxicity of the different adhesive systems and the inflamatory response of gingival cells exposed to these chemicals. Further investigations on these interesting subjects must be designed to achieve a standarized retention protocol that preserves the enamel surface and avoids the damaging of any dental structure. In this sense, the shape and amount of the 30-blade tungsten carbide bur is the main factor that contributes to the smoothnes of the enamel surface in comparison with the use of a white stone bur.

It has been suggested by in vitro study models that the optimal bond strengths for either composite pad (in fixed retainers) or orthodontic bracket bonding are in the range of 5.7–7.8 MPa, but it is difficult to apply these results to clinical practice [29]. In this study, the resistance to the debonding of the retention wire glued with composite resin pads to the teeth falls within the optimal range described above, i.e., it achieves optimal bond strength values. This is important to take into consideration when, during the controls after the orthodontic treatment, a force with a probe is applied to the retention wire to check if it is well bonded or needs to be fixed again with composite resin.

## 5. Conclusions

Disregarding the polishing bur, the debonded enamel surface showed increased roughness in comparison with the enamel of the control teeth. The polishing method does represent a difference in the enamel roughness after debonding.

The 30-blade tungsten carbide bur polishing provides a smoother surface, especially with a low speed. These results reject the null hypothesis. The bond strength between the tooth and the composite pad and retention wire was optimal for the purpose of a long-lasting retention of the teeth alignment. Further investigations for remineralization protocols are necessary given that all the methods studied caused an increase in the surface roughness of the treated enamel.

## Figures and Tables

**Figure 1 materials-16-02403-f001:**
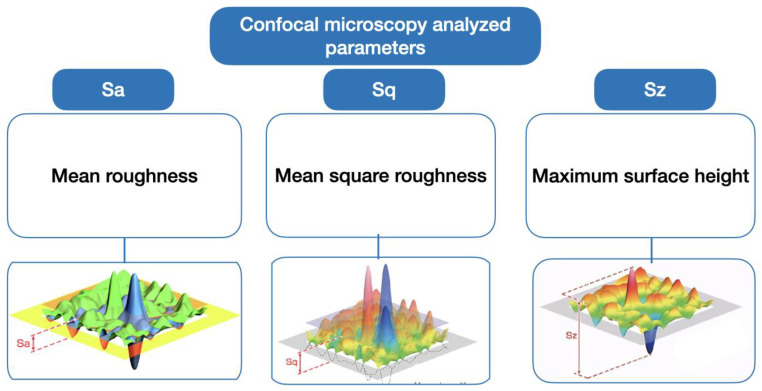
Scheme of confocal microscopy analyzed parameters: Sa (mean roughness), Sq (mean square roughness), and Sz (Maximum surface height).

**Figure 2 materials-16-02403-f002:**
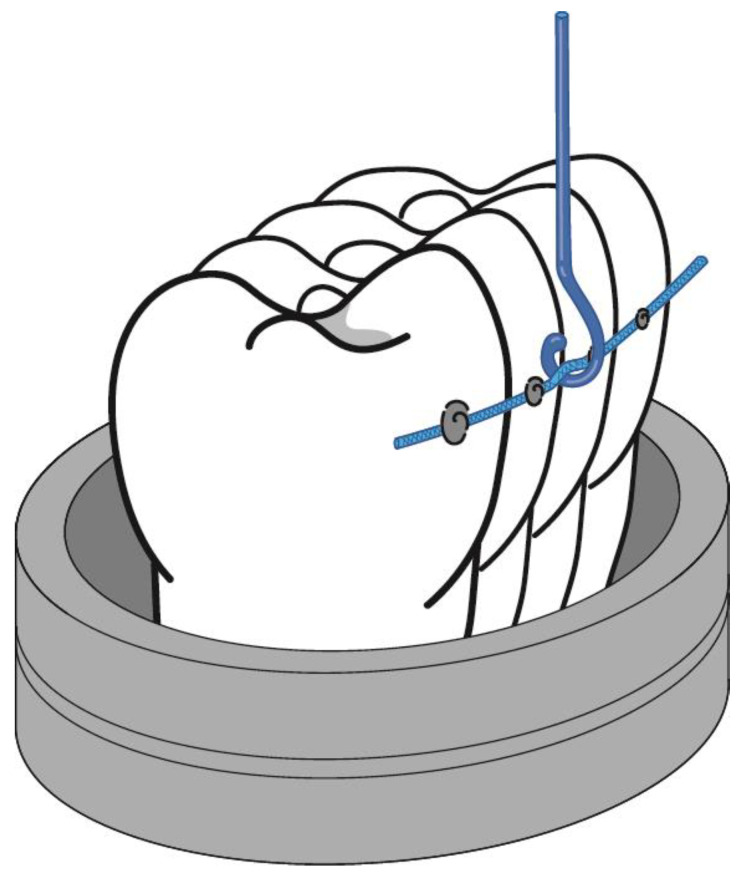
Scheme of mechanical tests.

**Figure 3 materials-16-02403-f003:**
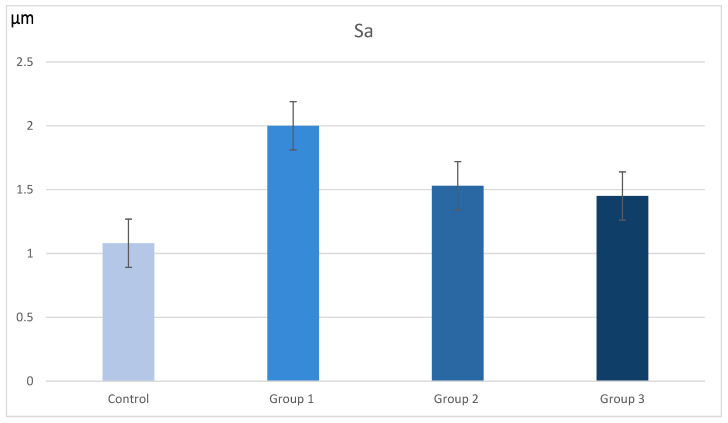
Bar diagram of the Sa variable values in μm (*p*-value < 0.001) for the three groups.

**Figure 4 materials-16-02403-f004:**
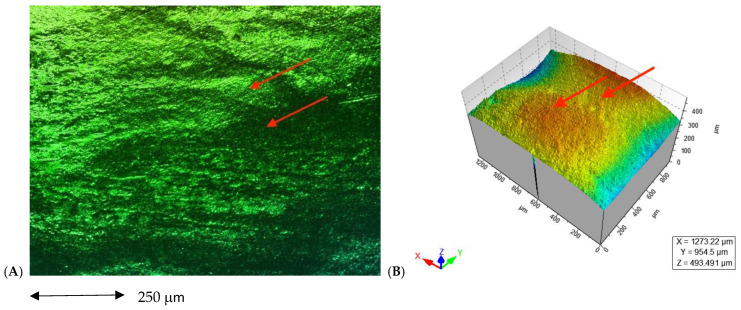
(**A**) Enamel appearance on confocal microscopy after debonding and polishing with a white stone bur high-speed handpiece (arrows point at the more representative indentations of the enamel surface); (**B**) 3D scheme of surface roughness of the selected representative sample for Group 1 (arrows point at the same area on the representative 3D model of the image’s roughness).

**Figure 5 materials-16-02403-f005:**
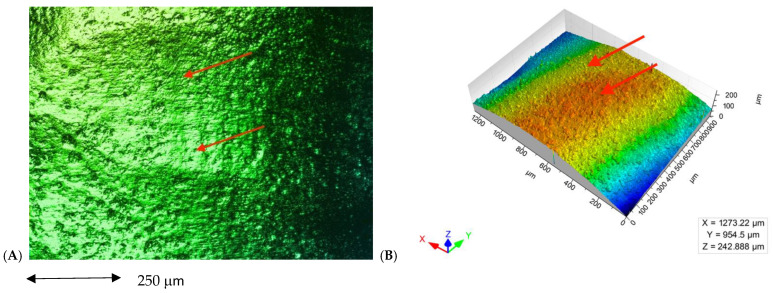
(**A**) Enamel appearance on confocal microscopy after debonding and polishing with a tungsten carbide bur high-speed handpiece (arrows point at the more representative indentations of the enamel surface); (**B**) 3D scheme of surface roughness of the selected representative sample for Group 2 (arrows point at the same area on the representative 3D model of the image’s roughness).

**Figure 6 materials-16-02403-f006:**
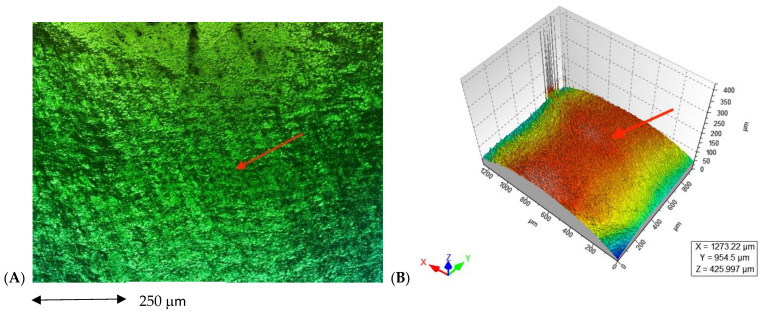
(**A**) Enamel appearance on confocal microscopy after debonding and polishing with a tungsten carbide bur low-speed handpiece (arrow points at only one area of enamel because it appears to have similar roughness on the entire surface); (**B**) 3D scheme of surface roughness of the selected representative sample for Group 3 (arrow points at the same area on the representative 3D model of the image´s roughness).

**Figure 7 materials-16-02403-f007:**
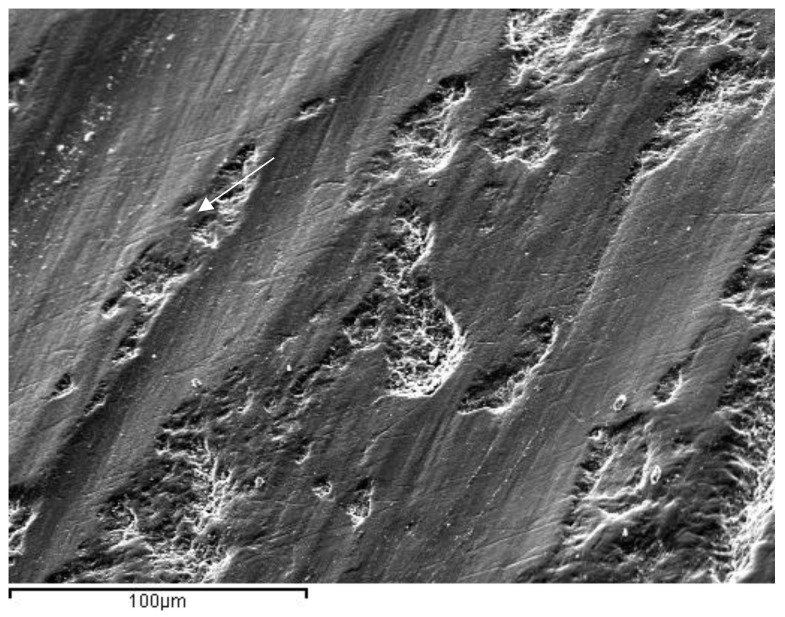
Representative image of enamel of a Group 1 sample after polishing with a white stone bur at a high speed (arrow points at the most evident indentation of the bur on the enamel surface).

**Figure 8 materials-16-02403-f008:**
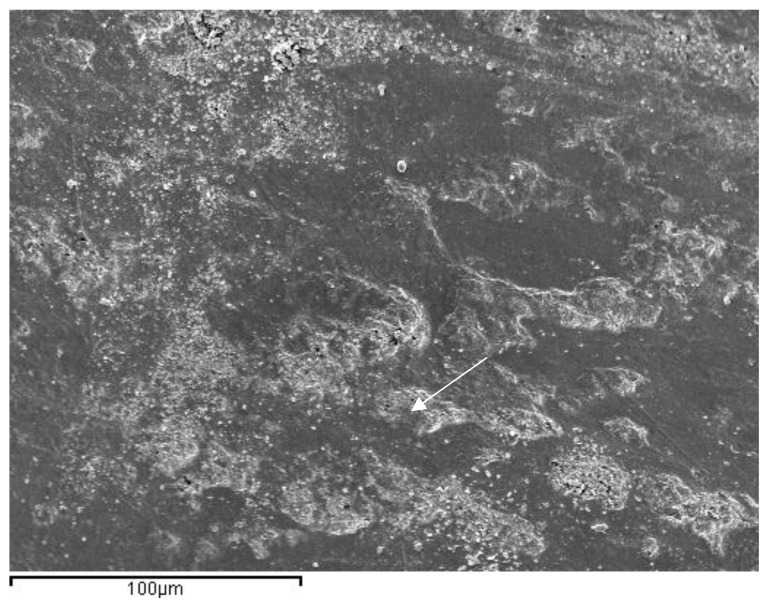
Representative image of enamel of a Group 2 sample after polishing with 30-blade carbide bur at a high speed (arrow points at the most evident indentation of the bur on the enamel surface).

**Figure 9 materials-16-02403-f009:**
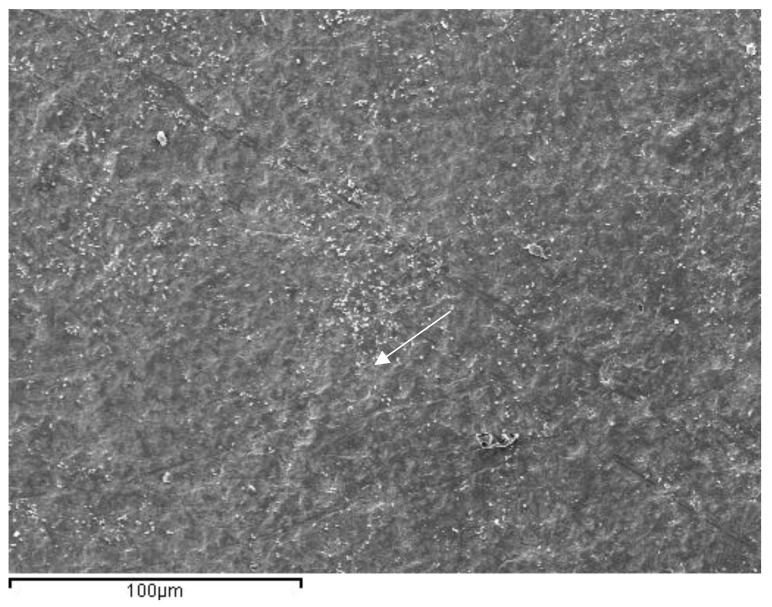
Representative image of enamel of a Group 3 sample after polishing with a 30-blade carbide bur low-speed handpiece (arrow points at the most evident indentation of the bur in the enamel surface).

**Table 1 materials-16-02403-t001:** Composition of fixed retention materials for all groups.

Material	Manufacturer	Components	Composition
Transbond XT	3MUnitek, Monrovia, CA, USA	Etching Gel Primer Paste	37% phosphoric acid, tetraethyleneglycol 1.39 dimethacrylate (TEGDMA), bisphenol-A-diglycidelmethacrylate (Bis-GMA); Bis-GMA, TEGDMA, silane-treatedquartz, amorphoussilica, camphorquinone
Gold Chain	Reliance Ortho Prod. Inc., Itasca, IL, USA	Gold Silver Copper Nickel Zinc	Solid metal alloy: gold (1.0–76.1%), silver (0.3–86.0%), copper (2.0–90.8%), nickel (0.0–22.0%), zinc (0.3–22.0%)

**Table 2 materials-16-02403-t002:** Polishing system of each group after debonding.

Group	Bur	Manufacturer
Group 1	Arkansas white stone bur high-speed handpiece (REF: FGAR661.030)	AXIS Dental Sàrl Ch. du Closalet 4 1023 Crissier Swizerland
Group 2	Tungsten carbide bur high-speed handpiece (REF: CA1S021)	AXIS Dental Sàrl Ch. du Closalet 4 1023 Crissier Swizerland
Group 3	Tungsten carbide bur low-speed handpiece (REF: FG44E018)	AXIS Dental Sàrl Ch. du Closalet 4 1023 Crissier Swizerland

**Table 3 materials-16-02403-t003:** Normality tests performed for each variable (Sa, Sq, Sz) and group at a 95% confidence interval. The red color stated for the statistically significant values.

	Differences: G1-Control		Differences: G2-Control		Differences: G3-Control	
	Dif (I.C)	*p*-Value	Dif (I.C)	*p*-Value	Dif (I.C)	*p*-Value
Sa	0.92 (0.53, 1.31)	<0.001	0.45 (0.24, 0.66)	<0.001	0.37 (0.13, 0.61)	0.005
Sq		0.001		0.124		0.469
Sz		0.885		0.357		0.357

**Table 4 materials-16-02403-t004:** Mean and median values of the three variables in µm.

	Control	Group 1	Group 2	Group 3
Sa (mean)	1.08 (0.08)	2.00 (0.69)	1.53 (0.35)	1.45 (0.41)
(median)	1.06 (1.02–1.12)	1.77 (1.53–2.14)	1.44 (1.30–1.65)	1.39 (1.21–1.61)
Sq (mean)	1.94 (0.13)	4.07 (2.02)	3.31 (1.98)	2.87 (1.66)
(median)	1.98 (1.90–2.01)	3.60 (2.84–4.13)	2.45 (2.00–4.39)	2.10 (1.83–3.51)
Sz (mean)	115.61 (8.47)	169.55 (122.40)	143.16 (144.26)	134.76 (15.91)
(median)	114.27 (109.26–120.63)	126.58 (72.22–252.94)	64.71 (34.22–290.26)	54.63 (42.82–266.13)

## Data Availability

The authors can provide details of the research upon request by letter and commenting on their needs.

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
