# Peer review of "Enamel Evaluation after Debonding of Fixed Retention and Polishing Treatment with Three Different Methods"

_materials, 2023, doi:10.3390/ma16062403_

Round 1
Reviewer 1 Report
1. Reduce the abstract size.
2. Add some history of dental composites and add the recommended article in the introduction.
a. Effect of aluminium oxide, titanium oxide, hydroxyapatite filled dental restorative composite materials on physico-mechanical properties
b. Ranking and selection of dental restorative composite materials using FAHP-FTOPSIS technique: an application of multi criteria decision making technique
3. Missing captions; Table 1 and 2 in Material and Methods section.
4. This is Materials and Methods not Material and Methods. Correct it
5. Tensile test was significant or not. Section 2.7; Provide any reference for Tensile test.
6. Figure 7, 8, and 9; a white stone bur, 30 blade carbide bur, 30 blade carbide bur. why did the authors take same polishing for Group 2 and 3.
7. Conclusions section; remove the points and make a single paragraph.
Author Response
REVIEWER 1
Dear Reviewer,
Thanks for taking the time to review our manuscript and suggest to us to improve our work by providing a lot more detail. We have done so, and we are now submitting a manuscript that not only addresses the points you specifically raised but also many others that we have considered in order to deliver what we think is a much improved version of our work. This version includes more paragraphs, English grammar revisions in all main sections, new references. Thanks a lot and happy new year. We are looking forward to your comments.
Sincerely,
Francisco-Javier Gil Mur
- Reduce the abstract size.
DONE
- Add some history of dental composites and add the recommended article in the introduction.
- Effect of aluminium oxide, titanium oxide, hydroxyapatite filled dental restorative composite materials on physico-mechanical properties
Dear reviewer, thank you for the suggestion of integrating this research in our study; however, we were unable to find this tittle in the data bases, if you could provide a reference for this topic, I am sure it would be of much interest for the study and we would be glad to add it into our references
- Ranking and selection of dental restorative composite materials using FAHP-FTOPSIS technique: an application of multi criteria decision making technique
Added
- Missing captions; Table 1 and 2 in Material and Methods section.
Done.
- This is Materials and Methods not Material and Methods. Correct it
Done
- Tensile test was significant or not. Section 2.7; Provide any reference for Tensile test.
Tests were performed to quantify the debonding force of the retention. New reference has been added.
- Figure 7, 8, and 9; a white stone bur, 30 blade carbide bur, 30 blade carbide bur. why did the authors take same polishing for Group 2 and 3.
Typing error in the description of the group already corrected
- Conclusions section; remove the points and make a single paragraph.
Done
Author Response
REVIEWER 2
Dear Reviewer,
Thanks for taking the time to review our manuscript and suggest to us to improve our work by providing a lot more detail. We have done so, and we are now submitting a manuscript that not only addresses the points you specifically raised but also many others that we have considered in order to deliver what we think is a much improved version of our work. This version includes more paragraphs, English grammar revisions in all main sections, new references. Thanks a lot and happy new year. We are looking forward to your comments.
Sincerely,
Francisco-Javier Gil Mur
The paper entitled “Enamel evaluation after debonding of fixed retention and polishing treatment with three different methods” is an original article reporting the effects different types of burs on the surface of the debonded enamel.
1.The manuscript requires revision for the language, which is not always correct in grammar and syntax.
Line 38-39 “For this reason, live long ( the term is long-life) retentions are more commonly advised every day by clinicians[1].”
Done
39-39 Line 51-53 “This is a critical moment, in which the selection of burs and rotary instruments to be used will affect the ability to remove the remnant materials while minimising the damage to the enamel structure.”(consider revising)
Revised and modified
51-53 2. The authors should consider to modify the following statement
:Line 72-72 “The aim of this study is to analyze different methods of enamel treatment after detachment of lingual retainers and observe the repercussions of this removal techniques on the enamel, in order to propose the most advisable removal protocol that can be standarized and widely used in the future;”
Modified
The study was conducted on buccal surface of the extracted premolars, so it is advisable to remain to the results obtained on these buccal surfaces, without extrapolation to the lingual surface (as the authors even suggested that might offer different results due to the lingual area, lines 252-269)
Done
3.”The scheme of the mechanical tests that can be seen in Figure 2” in line 168 , show the buccal position of the bracket, so you should remain at the buccal surface analysis.
All tests are performed on the buccal Surface to make comparisons with other studies. Figure 2 was changed with the proper scheme of the mechanical test.
4.Lines 312-314 There is no mentioning in the abstract or in the introduction about a secondary aim, as presented in the above lines “For this reason, the aim of this study was to identify the ideal sandblasting particle to establish a protocol that would improve the adhesion of the detached bracket to the tooth. From the results obtained, alumina-blasted brackets show a significant increase in adhesion.” Consider revising the abstract and the introduction part of the article.
These sentences have been corrected according to the reviewer.
- No.34. in references show no direct relation to the studied subject of the article. 34. Gil FJ, Espinar E, Llamas JM, Manero JM, Ginebra MP.Variation of the superelastic properties and nickel release from original 422 and reused NiTi orthodontic archwires. J.Mech.Beh.Biome. Mater. 2012; 6 : 113-119. DOI. 10.1016/j.jmbbm.2011.11.005. Consider removing from references.
I agree, the reference has been deleted.
Reviewer 3 Report
The aim of the paper is to analyze different methods of enamel polishing after detachment of orthodontic retainers. The topic is up-to-date and corresponds to the journal’s area. The findings in the research will be not only of high scientific interest but also of high interest of clinicians in dentistry.
The abstract is well structured and informative enough. The structure of the manuscript is well designed. The adequate investigation methods are used. The conclusions correspond to the results obtained. All figures and tables are cited in the text, however the reference [28] is not cited in the text. There are not missing tables, figures and references.
However, there are several remarks, which should be edited.
1. Materials and methods
a. Page 3, row 111 – “…pulling force” –did you control the pulling force?
b. 2.7. Mechanical tests – how did you calculate adhesion strength?
2. Results
a. Table 3 – It is not designated which parameter is investigated in the table.
b. Page 10, row 235 – “The value of bond strength and stress…” You investigated the adhesion/bond strength by tensile test. You did not investigate the stresses in the material by Finite Elements Analysis for example. So, you have to edit in the text the term “stress” with “strength”.
c. Figure 10 – the same “adhesion strength” instead of “stress”.
3. Discussion
a. Page 11, rows 247-249 – “According to Cardoso et al [16]. an ideal removal method must be of greater hardness than the adhesion material, but lesser than that of the enamel to avoid loss.” This sentence is not clear, please edit.
b. Page 12, rows 270-274 – the explanation about SEM is redundant.
c. The referemces are not cited in ascending order and the reference [28] is not cited in the text.
Author Response
REVIEWER 3
Dear Reviewer,
Thanks for taking the time to review our manuscript and suggest to us to improve our work by providing a lot more detail. We have done so, and we are now submitting a manuscript that not only addresses the points you specifically raised but also many others that we have considered in order to deliver what we think is a much improved version of our work. This version includes more paragraphs, English grammar revisions in all main sections, new references. Thanks a lot and happy new year. We are looking forward to your comments.
Sincerely,
Francisco-Javier Gil Mur
The aim of the paper is to analyze different methods of enamel polishing after detachment of orthodontic retainers. The topic is up-to-date and corresponds to the journal’s area. The findings in the research will be not only of high scientific interest but also of high interest of clinicians in dentistry.
The abstract is well structured and informative enough. The structure of the manuscript is well designed. The adequate investigation methods are used. The conclusions correspond to the results obtained. All figures and tables are cited in the text, however the reference [28] is not cited in the text. There are not missing tables, figures and references.
However, there are several remarks, which should be edited.
- Materials and methods
- Page 3, row 111 – “…pulling force” –did you control the pulling force?
The methodology of mechanical testing has been improved.
- 2.7. Mechanical tests – how did you calculate adhesion strength?
The authors have introduced this explanation.
- Results
- Table 3 – It is not designated which parameter is investigated in the table.
DONE
- Page 10, row 235 – “The value of bond strength and stress…” You investigated the adhesion/bond strength by tensile test. You did not investigate the stresses in the material by Finite Elements Analysis for example. So, you have to edit in the text the term “stress” with “strength”.
Corrected according to the reviewer. In materials and methods have been introduced the explanation about the determination of the adhesion strength.
- Figure 10 – the same “adhesion strength” instead of “stress”.
Done
- Discussion
- Page 11, rows 247-249 – “According to Cardoso et al [16]. an ideal removal method must be of greater hardness than the adhesion material, but lesser than that of the enamel to avoid loss.” This sentence is not clear, please edit.
Edited
- Page 12, rows 270-274 – the explanation about SEM is redundant.
Revised according to the reviewer
- The referemces are not cited in ascending order and the reference [28] is not cited in the text.
Revised.
Reviewer 4 Report
This study aimed to analyze different methods of enamel polishing after detachment of orthodontic retainers. Forty-five healthy premolars were divided into 3 groups according to the polishing bur after debonding, and four specimens without intervention were used as control. Comments are listed below,
1. I suggest to further refine the abstract to highlight the significance and innovation of the presented work.
2. What is the motivation and driver behind this study?
3. Since a large number of references are review, it is suggested to add a table for summarizing and comparison.
4. Please discuss the uncertainty factors relating to the sample preparation.
5. The format of Table 3 and 4 should be revised.
6. The main body of the article always describes the data, but rarely explains why the phenomenon occurs. Modifications are recommended.
7. The graphs throughout the paper are hard to read, and the illustrations are obscure.
Author Response
REVIEWER 4
Dear Reviewer,
Thanks for taking the time to review our manuscript and suggest to us to improve our work by providing a lot more detail. We have done so, and we are now submitting a manuscript that not only addresses the points you specifically raised but also many others that we have considered in order to deliver what we think is a much improved version of our work. This version includes more paragraphs, English grammar revisions in all main sections, new references. Thanks a lot and happy new year. We are looking forward to your comments.
Sincerely,
Francisco-Javier Gil Mur
This study aimed to analyze different methods of enamel polishing after detachment of orthodontic retainers. Forty-five healthy premolars were divided into 3 groups according to the polishing bur after debonding, and four specimens without intervention were used as control. Comments are listed below,
- I suggest to further refine the abstract to highlight the significance and innovation of the presented work.
Done
- What is the motivation and driver behind this study?
It is two-fold: 1. To check the quality of the refinement of the enamel after debonding in order not to damage it further. The enamel doesn´t have the capacity to regenerate, so all we can do to preserve as much as possible its integrity is important. 2. To quantify the debonding force. Again, once a retention is debonded the enamel should be polished and the retention bonded again to preserve the alignment. If the bonding strength is correct, the retention will remain in its position more time.
- Please discuss the uncertainty factors relating to the sample preparation.
The sample of premolars came from premolars extraction due to orthodontic purposes and with the approval of the Ethics Committee of our University.
- The format of Table 3 and 4 should be revised.
Revised according to the reviewer
- The main body of the article always describes the data, but rarely explains why the phenomenon occurs. Modifications are recommended.
Added according to the reviewer
- The graphs throughout the paper are hard to read, and the illustrations are obscure.
We agree with your opinion but it was not possible to enlarge the images of the Figure 1.
Round 2
Reviewer 4 Report
The author has replied my comments.